# High-Throughput Label-Free Isolation of Heterogeneous Circulating Tumor Cells and CTC Clusters from Non-Small-Cell Lung Cancer Patients

**DOI:** 10.3390/cancers12010127

**Published:** 2020-01-03

**Authors:** Mina Zeinali, Maggie Lee, Arthi Nadhan, Anvya Mathur, Casey Hedman, Eric Lin, Ramdane Harouaka, Max S. Wicha, Lili Zhao, Nallasivam Palanisamy, Mathias Hafner, Rishindra Reddy, Gregory P. Kalemkerian, Bryan J. Schneider, Khaled A. Hassan, Nithya Ramnath, Sunitha Nagrath

**Affiliations:** 1Chemical Engineering, University of Michigan, 2800 Plymouth Road, NCRC, Building 20-3rd Floor, Ann Arbor, MI 48109, USA; mzeinali@umich.edu (M.Z.); maggifly@umich.edu (M.L.); arthinad@umich.edu (A.N.); anvmat@umich.edu (A.M.); gogodidi@umich.edu (E.L.); 2Biointerfaces Institute, University of Michigan, 2800 Plymouth Road, NCRC B10-A184, Ann Arbor, MI 48109, USA; 3Institute for Medical Technology of Heidelberg University & University of Applied Sciences Mannheim, Paul-Wittsack-Straße 10, 68163 Mannheim, Germany; m.hafner@hs-mannheim.de; 4Molecular, Cellular, and Developmental Biology, University of Michigan, 1105 North University Avenue, 2220 Biological Science Building, Ann Arbor, MI 48109, USA; caseych@umich.edu; 5Department of Internal Medicine, University of Michigan, 1500 E Medical Center Dr, Ann Arbor, MI 48109, USA; rharouaka@gmail.com (R.H.); mwicha@umich.edu (M.S.W.); kalemker@umich.edu (G.P.K.); bryansch@umich.edu (B.J.S.); khaledh@umich.edu (K.A.H.); 6Biostatistics Department, University of Michigan, Ann Arbor, MI 48109, USA; zhaolili@umich.edu; 7Department of Urology, Henry Ford Health System, 1 Ford Place, Room 2D26, Detroit, MI 48202, USA; npalani1@hfhs.org; 8Department of Surgery, University of Michigan, 1500 E Medical Center Dr, Ann Arbor, MI 48109, USA; reddyrm@med.umich.edu; 9Veterans Administration Ann Arbor Healthcare System, 2215 Fuller Road, Ann Arbor, MI 48105, USA

**Keywords:** circulating tumor cells (CTCs), CTC clusters, non-small-cell lung cancer (NSCLC), inertial microfluidics, epithelial-to-mesenchymal transition (EMT)

## Abstract

(1) Background: Circulating tumor cell (CTC) clusters are emerging as clinically significant harbingers of metastases in solid organ cancers. Prior to engaging these CTC clusters in animal models of metastases, it is imperative for technology to identify them with high sensitivity. These clusters often present heterogeneous surface markers and current methods for isolation of clusters may fall short. (2) Methods: We applied an inertial microfluidic Labyrinth device for high-throughput, biomarker-independent, size-based isolation of CTCs/CTC clusters from patients with metastatic non-small-cell lung cancer (NSCLC). (3) Results: Using Labyrinth, CTCs (PanCK+/DAPI+/CD45−) were isolated from patients (*n* = 25). Heterogeneous CTC populations, including CTCs expressing epithelial (EpCAM), mesenchymal (Vimentin) or both markers were detected. CTCs were isolated from 100% of patients (417 ± 1023 CTCs/mL). EpCAM− CTCs were significantly greater than EpCAM+ CTCs. Cell clusters of ≥2 CTCs were observed in 96% of patients—of which, 75% were EpCAM−. CTCs revealed identical genetic aberrations as the primary tumor for *RET*, *ROS1* , and *ALK* genes using fluorescence in situ hybridization (FISH) analysis. (4) Conclusions: The Labyrinth device recovered heterogeneous CTCs in 100% and CTC clusters in 96% of patients with metastatic NSCLC. The majority of recovered CTCs/clusters were EpCAM−, suggesting that these would have been missed using traditional antibody-based capture methods.

## 1. Introduction

Cancer metastasis is the cause of cancer death in the vast majority of patients with solid organ cancers, including lung cancer [1]. Circulating tumor cells (CTCs) represent a heterogeneous population of malignant cells that disseminate from tumors into the bloodstream, as single cells or tumor cell clusters, enabling their transportation to distant sites and the seeding of metastatic tumors [2]. Despite decades of research, therapies to reduce and prevent metastases have mostly been elusive. CTCs have been shown to correlate with incidence of metastasis after initial surgery and the numbers of CTCs correlate with progression-free and overall survival in many cancers, including lung cancer [3,4,5].

CTC clusters are defined as groups of two or more aggregated CTCs, often consists of other non-tumor cells including endothelial cells, erythrocytes, stromal cells, leukocytes, platelets, and cancer associated fibroblasts found in the blood of patients with solid tumors [6]. This cocktail of cells provides a local microenvironment for protection from death in the circulation by minimizing shear stress, immune system attack and through facilitating colonization [7]. Although CTC clusters are even rarer than single CTCs (about 3% of all CTC) [8], they have been observed in various cancers including lung cancer [9]. More recently technological advances have allowed researchers to capture CTC clusters. Several groups are studying these clusters as potential surrogates for tumor biology and aggressiveness. We have previously demonstrated the presence of CTC clusters in patients with early-stage lung cancer [4]. We have chosen here to focus our studies on the detection of these CTC clusters in metastatic NSCLC.

Lung cancer is among the most prevalent forms of cancer and has one of the highest mortality rates, with a 5 year survival rate of only 19% [10,11]. About 85% of lung cancers are classified as non-small-cell lung cancer (NSCLC) [12]. Less than 30% of patients with NSCLC present with earlier stages, allowing curative surgical resection [11]. The majority of patients present with locally advanced or metastatic cancer. Despite improvements in targeted and immunotherapy, the overall survival is dismal with less than 25% of all NSCLC patients living 5 years or longer. There are several unique characteristics that contribute to the aggressiveness of NSCLC including late detection, high degree of tumor heterogeneity and resistance to available therapies. There is an urgent, unmet need to identify ways to improve earlier diagnosis/recurrence, capture tumor heterogeneity and understand why certain lung cancers metastasize to specific organs such as the brain. Downstream molecular studies using CTC clusters and performing in vivo experiments may give the answers to these questions. These CTC clusters are characterized by the heterogeneity of surface markers just as individual CTCs. It is therefore imperative to develop a platform that could enable high fidelity capture of these CTC clusters. 

The detection of CTC clusters is even more challenging due to the rarity among other CTCs, and possible dissociation of clusters during the blood sample processing [6]. Therefore, high-throughput capability of isolation based on shape, size, composition, and label-free with minimum manipulation would be an efficient platform to isolate CTC clusters [6].

CTC enrichment technologies utilize either biological (label-dependent or immunoaffinity-based) or physical (label-independent or size-based) properties of cells [2,13,14,15,16,17,18,19]. Currently, available CTC technologies more often depend on surface markers, such as EpCAM, to identify these cells, potentially missing a large population that may be more “stem-like” in nature [20]. On the other hand, label-free technologies, depending on size, density, and dielectrophoresis properties, isolate heterogeneous population of CTCs. There are other label-free technologies available for CTC detection, for instance, EPISPOT, which discriminates between viable and apoptotic CTCs using protein secretion [21], FISH-based assays [22,23], and metabolic approaches that measure the fluctuations in pH or lactate concentration in CTCs [24]. Label-free technologies are often limited by the throughput. Hence, a high-throughput microfluidic-based biomarker-independent CTC isolation technology can address the current challenges [25].

CTC clusters were observed form patient blood samples using different technologies including density-based cell separation apparatus (AccuCyte) (presence of clusters) [26], Epic (5.8 CTC clusters/mL from metastatic NSCLC patient samples) [27], Vortex Chip (presence of clusters) [28], lithographic microfilters (presence of clusters) [29], ScreenCell (presence of clusters) [30], Cluster Chip (30%–40% of patients ranging from 0.1–0.5 clusters/mL) [31], and flexible micro spring array (FMSA) (30% of NSCLC patient samples) [32]. A detailed review of CTC cluster capture platforms was performed by Rostami et al [6]. Herein, we present a technological advance using a high-throughput, label-independent, size-based inertial microfluidic separation device, Labyrinth, previously used to detect CTCs in breast and pancreatic cancer patients [25]. Both inertial focusing and Dean flow were incorporated for size-based separation, enabling the continuous focusing of all cells while separating CTCs from smaller blood cells at the outlet [25]. However, the main disadvantage of size-based separation methods is the loss of smaller CTCs, which may result in losing valuable information from the patient [33,34].

## 2. Results

### 2.1. Optimization of Labyrinth for Cell Recovery

The microfluidic Labyrinth device consists of a long curvature path (as shown in Figure 1A by loading Labyrinth with red dye to delineate the device’s structure), an inlet through which the sample is introduced and four outlet channels. Typically, most of the white blood cells (WBCs) are collected through outlet 1, whereas CTCs from outlet #2 (Figure 1A).

The device was optimized and tested for inertial separation of cancer cells using human lung cancer cell line, H1650. To demonstrate cell focusing in the device, the same concentration of pre-labeled cancer cells with CellTracker Green and pre-labeled WBCs with DAPI (4′,6-diamidino-2-phenylindole) (each 1000 cells/mL) were spiked into phosphate-buffered saline (PBS) and separated using the Labyrinth. Different flow rates ranging from 2300–2500 µL/min were tested for further optimization. Products were collected after flow stabilization (1.5 min) and the percentage of CTC recovery and WBC depletion were determined by cell counting. Using the H1650 cell line, 82% ± 5% of cancer cells were recovered at a flow rate of 2500 µL/min (Figure 1B), while 78% ± 18% of WBCs were removed (Figure 1C).

To identify CTCs in patient samples, immunofluorescence (IF) staining was optimized with a panel of antibodies (anti-human CD45 (cluster of differentiation 45), anti-human PanCK (pan-Cytokeratin), anti-human EpCAM (Epithelial cell adhesion molecule), and anti-human Vimentin) using lung cancer cell lines including H1975 and A549. We were able to show that both H1975 and A549 were positive for PanCK and negative for CD45, whereas these cells expressed EpCAM and Vimentin differentially as expected (Figure 1D).

### 2.2. Isolation of CTCs from NSCLC Patients

Peripheral blood samples were collected from 25 patients with metastatic, stage IV NSCLC (Appendix A). Among these 25 patients, there were patients with *EGFR* mutations (*n* = 15), *ROS1* rearrangements (*n* = 6), and *ALK* fusion (*n* = 4). These patient samples were processed through the Labyrinth (as described in the Materials and Methods).

After CTC isolation, the product from outlet #2 was analyzed for CTCs. CTCs were detected by IF staining as described in the Materials and Methods. Cells with the PanCK+/CD45−/DAPI+ phenotype were identified and enumerated as CTCs. Figure 2A shows IF staining of an isolated single CTC stained positive for PanCK (red) and negative for CD45 (green) to distinguish CTCs from WBCs. Figure 2B,C illustrate confocal images of isolated CTCs in clusters of two (Figure 2B) and three cells (Figure 2C). We determined that all 25 patients (100%) had detectable CTCs with an average of 417 CTC/mL (10.2–5068) (Figure 2D). In contrast, low numbers of CTC/mL (0–3) were observed in the healthy controls (HCs, *n* = 3) (*p* = 0.0006) (Figure 2D and Appendix A) (Appendix A).

### 2.3. Identification of Heterogeneous Subpopulations of CTCs Isolated Using Labyrinth

Isolated CTCs from a subset of NSCLC patients (*n* = 23) were further examined to determine the percentage of cells that displayed epithelial and mesenchymal markers (Appendix A). We used EpCAM (for epithelial phenotype) and Vimentin (for mesenchymal phenotype), in addition to PanCK (tumor marker), CD45 (leukocyte marker), and DAPI (nuclear marker). CTCs were defined as cells positive for PanCK, and DAPI, but negative for CD45. We further categorized CTCs into subpopulations based on expression of EpCAM, Vimentin, or double expression of EpCAM and Vimentin (Figure 3A).

Figure 3B,C demonstrate the percentage of CTCs expressing EpCAM (EpCAM+/− in dark/light orange, respectively) and CTCs expressing Vimentin (Vimentin+/− in dark/light pink, respectively) from each patient. Of 23 patients, 17 demonstrated higher numbers of EpCAM− CTCs than EpCAM+ CTCs and six demonstrated higher numbers of EpCAM+ CTCs than EpCAM− CTCs. Of the captured CTCs among all patient samples (*n* = 23), 31% (96 CTCs/mL) were EpCAM+ CTCs, whereas 69% (336 CTCs/mL) were EpCAM− CTCs (*p* = 0.01) (Figure 3B and Appendix A), 45% (346 CTCs/mL) were Vimentin+ CTCs and 55% (85 CTCs/mL) were Vimentin− CTCs (Figure 3C and Appendix A).

### 2.4. CTC Clusters Isolation by Using Labyrinth

To investigate the ability of Labyrinth to isolate CTC clusters, we analyzed the CTCs isolated from the above cohort of patients for the presence of clusters (Figure 4A) (Appendix A). In addition to single CTCs, a large number of CTCs were observed in cluster forms (2 to 8 CTCs/cluster). On average, 31 ± 26 CTCs/mL of the recovered CTCs were in the form of single CTCs, whereas 386 ± 1025 CTCs/mL were in cluster forms (*p* = 0.001) (Figure 4B,C). The majority of patients (24/25) presented CTC clusters. Of the 24 patients with CTC clusters, 23 had ≥2 clusters. The majority (*n* = 22) presented with at least three CTC clusters. There were others with four clusters (*n* = 11), five clusters (*n* = 8), six clusters (*n* = 3), seven clusters (*n* = 3) and eight clusters (*n* = 1) (Figure 4D). Significantly higher numbers of recovered clusters from Labyrinth were EpCAM− *(p* = 0.005) (Appendix A) and 41% of the recovered CTC clusters displayed mesenchymal marker Vimentin (*p* = 0.007) (Figure 4E), suggestive of a preponderance of an epithelial-to-mesenchymal transition (EMT) phenotype in CTC clusters [7]. Based on our results, the group of patients who had a higher number of CTC clusters than single CTCs showed worse progression-free survival (PFS) (25 vs. 51 months *p* = 0.05) (Figure 4F). However, the difference was not statistically significant.

### 2.5. Genomic Analysis of Recovered CTCs Using Fluorescence In Situ Hybridization (FISH) Analysis

To investigate whether CTCs isolated by Labyrinth carry the genomic signature of the primary tumor, we analyzed CTCs from selected patients for genomic alterations in *ROS1*, *ALK*, and *RET* by performing interphase FISH as previously described [35].

A 5′ deletion of *ROS1* was detected by FISH in recovered CTCs from a patient with a known *ROS1* rearrangement in the primary tumor (P 03) (Figure 5A). Similarly, an *ALK* fusion was noted in the CTCs, matching the patient’s primary tumor (P 17) (Figure 5B), and a 3′ deletion of *RET* was found in recovered CTCs from a patient with a *RET* rearranged tumor (P 26) (Figure 5C). 

The total numbers of CTCs/mL in all patients with respect to their *EGFR*/*ROS1*/*ALK* status are shown in Figure 5D. Patients with *EGFR* mutations (*n* = 15) had an average 559 CTCs/mL (10–5068), those with *ROS1* rearrangements (*n* = 6) had an average of 220 CTCs/mL (15–551), and those with *ALK* fusion (*n* = 4) had an average of 178 CTCs/mL (70–363). However, the differences in the number of CTCs according to genotype were not statistically significant. 

## 3. Discussion

The field of study involving circulating tumor cells (CTCs) as well as circulating tumor DNA (ctDNA) has evolved at a rapid pace that may eventually provide a valuable point of care assessment that could be used to both predict the effectiveness of specific targeted therapies as well as prognosticate cancer outcomes respectively. Specifically, ctDNA is an established test for example for *EGFR* mutant lung adenocarcinoma to both diagnose and follow patients being treated with targeted therapies. CTCs on the other hand may be used in the future to reflect both the genotype and the phenotype of the underlying cancer and may serve as an ex vivo predictor of targeted drug sensitivity, using expanded CTCs. Therefore, CTCs may serve as a valuable adjunct to blood-based ctDNA that is currently in advanced stages of development for the detection of specific molecular subtypes of NSCLC. CTC clusters are rarer than single CTCs [8] and they have a short half-life of less than 2.5 h in circulation [36], therefore, isolation techniques must be sufficiently sensitive to identify, enumerate, and isolate intact CTC clusters in a reasonable time with minimal manipulation.

Current methods for isolation of CTCs and CTC clusters mostly rely on biomarker-dependent antibody-based capture. CTC clusters demonstrate a hybrid of epithelial–mesenchymal character. Moreover, multicellular clusters have a much smaller area-to-volume ratio thus minimizing antibody-binding regions [8]. Therefore, current isolation techniques, can underestimate CTC/clusters populations that may be stem-like or mesenchymal in phenotype, hampering detecting of heterogeneous populations of cancer cells. 

Hence, biomarker-independent technologies are eventually taking the lead in the field. By removing the bias of using molecular markers for CTC/clusters isolation, heterogeneous CTC/cluster subpopulations can be recovered using label-free isolation. Each subpopulation may carry distinct, important information that could predict patient outcomes. 

As demonstrated previously, Labyrinth incorporated both inertial focusing and Dean flow for size-based separation, enabling the continuous focusing of all cells while separating CTCs from smaller blood cells at the outlet [25]. In summary, the Labyrinth device employs a combination of long curvature structure with loops and sharp corners to enhance differential focusing, allowing ease of separation of CTCs and white blood cells (WBCs) from patient blood samples [25]. The loops have a small curvature ratio to provide enough channel length to achieve the total focusing of CTCs from other blood cells. In contrast, the sharp corners have a high curvature ratio to enhance the focusing of blood cells. The major difference between Labyrinth and all other spiral designs lies in the numerous corners placed across the flow pattern. It is proposed by Sun et al. that the sharp change in flow direction can increase the focusing of smaller particles (e.g., WBCs) [37]. 

We determined that 100% of processed NSCLC patient samples (*n* = 25) had detectable CTCs, with an average of 417 CTC/mL with the number of CTCs/mL ≥10 in 100%, ≥50 in 80%, ≥100 in 64%, ≥200 in 32%, ≥400 in 12%, and ≥1000 in 8% of NSCLC patient samples. Furthermore, of the recovered CTCs, a significantly higher number of EpCAM− compared to EpCAM+ CTCs were detected (*p* = 0.01). As such, CTC detection based only on EpCAM (EpCAM+) might have missed a substantial number of tumor cells [20,38]. 

High numbers of CTCs have been reported by others previously [39]. It is not unusual to observe a high number of CTCs in metastatic lung cancer patients [27,40]. Moreover, the numbers of CTCs do vary with the treatment response or resistance [41,42,43,44]. Hence, based on our healthy controls and also from the downstream genetic analysis confirming that CTCs presenting matched molecular signatures to the primary tumor, we are confident that the CTCs we observe in patients are clinically relevant.

We have monitored CTC counts on therapy, and we have observed that CTC counts match patient’s outcomes. For example, ten consecutive samples from patient P 14 were collected during different treatment regimens. The numbers of CTCs were analyzed during follow-up visits (Figure 6). This patient enrolled on this study following progression of her cancer after initially receiving four cycles of adjuvant chemotherapy with cisplatin/gemcitabine. She was noted to have an *EGFR* mutation and subsequently underwent therapy with erlotinib, afatinib, osimertinib (various tyrosine kinase inhibitors due to tolerability of therapy). Following further progression, she received chemotherapy with carboplatin/pemetrexed as well as radiation to brain metastasis. She was then placed on chemotherapy with pemetrexed. Higher numbers of CTCs were observed in her 2nd and 3rd visits compared to 1st visit (reflecting progression of her cancer on therapy). After 183 days (visits 3–4), her brain MRI showed an interval increase in the size of lesions at the junction of L frontal/parietal lobes and a new 5 mm lesion in L posterior fossa. She underwent stereotactic radiation to the brain lesions. However, at 238 days, she continued to show further progression in the brain. Her CTC counts increased at these visits compared to the previous follow-up visits. After 400 days the brain MRI showed response to treatment. Mild reduction in the size of known supra and infratentorial metastatic lesions and reduced degree of associated edema was observed. Her CTC counts were dropped drastically. After 425 days CT on chest showed subtle growth of several lung nodules consistent with metastatic disease progression. Her CTC counts increased again, in keeping with this thoracic progression and the counts continued to increase until she was placed on pemetrexed chemotherapy, when the CTC counts dropped. In summary, according to the available clinical information, the CTC numbers tracked the patient’s outcome over the course of treatment. 

It has been reported that the CTC clusters have a high propensity for survival in circulation and in the development of metastases [9]. Using the Labyrinth, we were able to not only isolate CTCs in a single form, but also in clusters (*p* = 0.001). Patients who had a higher number of CTC clusters than single CTCs had worse PFS. However, the observation was not significant in this cohort which could be due to the limited sample size (*p* = 0.05). We believe that these CTC clusters may provide an important tool to be used in mice studies (CTC cluster derived xenografts) to understand key biological processes such as metastases to specific organs. Additionally, these CTC clusters can be used to develop organoid models that can serve as a platform for ex vivo expansion and drug. The latter application can assist the clinician in identifying the “best in class” tyrosine kinase inhibitor (TKI) against a specific target. 

The discovery and development of small molecule TKIs has revolutionized therapy for specific molecular subsets of NSCLC patients [45]. However, the majority of patients will develop resistance to TKIs within a few months/years [45,46]. The resistance is often mediated by a secondary mutation in the target gene or alternative pathways that supervene and bypass the original signaling pathway [47]. Presently, the gold standard to ascertain mechanism of resistance is to perform a re-biopsy [48,49]. Real-time PCR (qPCR) [50], digital PCR (dPCR) [51], and NGS (Next-Generation Sequencing) technologies have been incorporated in detecting ctDNA, which is becoming a reliable alternative to tissue biopsies. Once the resistance mechanism is identified, there are newer therapies available to target the resistance mechanism. However, there is no definitive method to identify which one of the various drugs that target the “resistance causing mutation” might be the most effective for an individual patient [47]. There is, therefore, an unmet need, not only to identify resistance early and without a tissue biopsy, but also for drug sensitivity assays that predict response ex vivo. CTC/CTC cluster isolation as a non-invasive method could help to identify primary and secondary resistance mutations and allow for ex vivo drug testing to direct therapy specific to the patient.

We hereby present a novel platform, the Labyrinth that is not only highly sensitive for CTC and CTC cluster isolation, but also highly specific in detecting specific oncogenic driver mutations, reflective of the underlying cancer. Furthermore, CTC numbers measured CTCs over the course of therapy reflected patient clinical outcomes. We will focus our next set of studies in expanding these CTCs/clusters to develop ex vivo drug testing platforms.

## 4. Materials and Methods

### 4.1. Microfabrication

The fabrication of Labyrinth devices includes the design of film mask, the fabrication of SU-8 mold in clean room, and the fabrication of poly-dimethyl siloxane (PDMS) (Ellsworth Adhesives, Germantown, WI, USA) device in laboratory. The film mask was designed in laboratory using L-Edit software. The design was converted to a photomask (FineLine Imaging, Colorado Springs, CO, USA) and used to prepare a mold by traditional photolithography. In short, a negative photoresist SU-8 100 was utilized to form the mold for the PDMS device by applying soft lithography. Using a spin-coater, the negative photoresist layer, SU-8 100, was deposited onto silicon wafer with 2450 rpm rotating for 1 min. The wafer was then soft-baked for 10 min at 65 °C and 70 min at 95 °C. The mask was aligned to the wafer and is exposed to UV light for 20 s. Post-exposure-baking was applied for 3 min at 65 °C and 10 min at 95 °C. Then, the wafer was developed by soaking in developer for 6 min and in isopropyl alcohol (IPA) for 1 min to remove the inactivated photoresist. It was hard baked for 3 min at 150 °C. The height of the mold built on silicon wafer was ideally 100 μm, and the width of the channel was 500 μm.

To prepare the PDMS device from fabricated mold, 30 mL Sylgard polymer base and 3 mL curing agent were mixed well and poured onto silicon mold. The mixture was left in desiccator for 2 h to remove trapped air within the mixture. It was then heated at 65 °C overnight to harden the polymer. The polymer was cut into the desired shape, e.g., rectangular for Labyrinth, and punched with a needle for tubing later. The PDMS device was then bonded to standard-sized glass slides via plasma surface activation of oxygen. The bonded device was tubed with 0.66 mm diameter tubes.

### 4.2. Labyrinth Device

The design of the Labyrinth device was inspired by the Labyrinth in Greek mythology [25]. The device consists of 11 long loops and 56 sharp corners, with a total channel length of 637 mm, at a width of 500 µm and height of 100 µm. The device has one inlet and 4 outlet channels. The width of outlet #1 is 135 µm, outlet #2 is 180 µm, outlet #3 is 135 µm, and outlet #4 is 400 µm. The outlets were designed such that outlet #1 collects WBCs, outlet #2 collects CTCs, and outlets #3 and #4 collect other blood components.

### 4.3. Cell Preparation

Human lung cancer cell lines H1650, A549, and H1975 (ATCC) were cultured in RPMI 1640 supplemented with 10% Fetal Bovine Serum (FBS) and 1% Penicillin-Streptomycin (Invitrogen, Carlsbad, CA, USA) and maintained in an incubator at 37 °C in 5% CO_2_ and 95% relative humidity. Upon reaching >75%–80% confluence, the cells were harvested with 0.05% trypsin-EDTA (Ethylenediaminetetraacetic acid) (Invitrogen, Carlsbad, CA, USA). Prior to characterization experiments, cells were labeled with CellTracker™ Green CMFDA Dye (Invitrogen, Carlsbad, CA, USA). The density of cell suspensions was counted using a hemocytometer and spiked into phosphate-buffered saline (PBS) or healthy control (HC) blood for analysis of CTCs recovery from Labyrinth.

### 4.4. Characterization Experiments for Labyrinth (Cell Lines)

The Labyrinth was primed with 1% Pluronic acid solution (diluted in 1× PBS) at a flow rate of 100 μL/min for 10 min and then incubated for 10 min to prevent cell adhesion to channel walls. To demonstrate cell specificity using the Labyrinth, the same number of pre-labeled cancer cells (from cell lines) and DAPI-labeled WBCs (each 1000 cells/mL) were mixed and spiked into PBS and flowed through the device at different flow rates (2300, 2400, 2500 µL/min). After allowing for stabilizing of flow streams (1.5 min) within the Labyrinth, products from outlet #2 were collected for cell counting to calculate the recovery and percent of depleted WBCs. Images were taken using a Nikon Eclipse Ti fluorescence microscope under FITC and DAPI filters to categorize and document cancer cells in isolation from other blood components.

### 4.5. Experimental Protocol for Labyrinth (Patient Samples)

The experimental protocol was approved by the Ethics committee (Institutional Review Board and Scientific Review Committee) of the University of Michigan. All patients and healthy blood donors gave their informed consent to participate in the study (HUM00119934). The patients had to have a diagnosis of driver mutant lung adenocarcinoma for whom a targeted therapy option was available. 

Briefly, 5 mL of blood was collected in EDTA tubes and processed through the Labyrinth within 2 h of collection. RBCs in the blood samples were removed using density separation with dextran solution (6% dextran solution, M.W. 250,000) prior to processing in the Labyrinth. However, there was no centrifugation step prior to sample processing. The blood sample mixed with dextran solution was kept at room temperature for 1 h to allow the RBCs to settle, aided by density difference. The supernatant, which included all whole blood components except RBCs, was carefully removed and diluted with PBS (1:3). The diluted dextran supernatant was then processed through the Labyrinth at a flow rate of 2500 μL/min. The product from outlet #2 was collected after stabilization. 

### 4.6. Immunofluorescence (IF) Staining

The product from outlet #2 was processed using a Thermo Scientific TM Cytospin Cytocentrifuge (Kalamazoo, MI, USA). A poly-lysine coated slide was placed into the cytospin funnel and 250 μL of sample was added to each cytospin funnel and cytocentrifuged at a speed of 800 rpm for 10 min. Samples were fixed on the cytoslides using 4% paraformaldehyde (PFA) and cytocentrifuged at the same conditions as described above. Slide samples were permeabilized by applying 0.2% Triton X-100 solution for 3 min. Slides were then washed with PBS (X 3) for 5 min and blocked using 10% donkey serum for 30 min at room temperature. 

The panel of antibodies (mouse anti-human CD45 (IgG2a, Biorad, Hercules, CA, USA), mouse anti-human PanCK (IgG1 Biorad, Hercules, CA, USA), anti-human EpCAM, biotinylated (goat IgG, R&D systems, Minneapolis, MN, USA), and rabbit anti-human Vimentin (Abcam, Cambridge, MA, USA) were optimized using lung cancer cell lines including H1975 and A549. The slides were then incubated with a cocktail of primary antibodies (anti-CD45, anti-PanCK, anti-EpCAM, and anti-Vimentin) overnight at 4 °C, followed by PBS wash (X 3) for 5 min the following day. Slides were incubated in the dark with secondary antibodies (Alexa Fluor 488, 546, 750, and 647) for 1.5 h at room temperature. Finally, slides were washed with PBS (X 3) for 5 min and mounted using Prolong Gold Antifade Mountant with DAPI. The immunofluorescently stained slides were imaged using a Nikon TI fluorescent microscope at 20X magnification for enumeration.

### 4.7. Fluorescence In Situ Hybridization (FISH) Analysis

Bacterial artificial chromosomes (BACs) were obtained from the BACPAC Resource Center (Oakland, CA, USA) for the preparation of probes. For the detection of gene rearrangements, deletion or amplification, the following probes were used: for *ROS1*, RP11-1110L9 (5′ to *ROS1*) and RP11-605K7 (3′ to *ROS1*), for *RET*, RP11-124O11 (5′ to *RET*) and RP11-718J13 (3′ to *RET*), for *EGFR*, RP11-23F4 (5′ to *EGFR*) and RP11-56P1 (3′ to *EGFR*), and for *ALK*, RP11-993C21 (5′ to *ALK*) and RP11-984I21 (3′ to *ALK*). The integrity and accurate localization of all probes were verified by hybridization to metaphase spreads prepared from normal peripheral lymphocytes. Slides were examined using a Zeiss Axioplan 2 microscope equipped with image processing software ISIS (Metasystems). FISH signals were scored manually under 100× oil immersion in morphologically intact and non-overlapping nuclei and greater than 100 cells from each slide were recorded.

### 4.8. Statistical Analysis

All results are presented as the mean ± standard deviation. The Mann–Whitney test (two-tailed) was used to compare the differences between total numbers of CTCs in patient samples (*n* = 25) and healthy controls (*n* = 3). Wilcoxon test were used to compare EpCAM+ and EpCAM− CTCs, Vimentin+ and Vimentin− CTCs, as well as single CTC vs. CTC clusters. Mann–Whitney unpaired t-test analysis was used for comparing patient cohort (*n* = 25) vs. healthy controls (*n* = 3). Log-rank (Mantel–Cox) tests were used to analyze the Kaplan–Meier progression-free survival (PFS) graph of CTC clusters vs. single CTCs across patient samples (*n* = 24). Statistical significance was defined as a two-sided *p* < 0.05. Analyses were conducted using GraphPad Prism (San Diego, CA, USA). 

## 5. Conclusions

The present study was devised to evaluate the detection of heterogeneous CTC/cluster populations isolated from metastatic NSCLC patient blood samples using a high-throughput, label-free, microfluidic Labyrinth device. CTCs were isolated from these patients whose lung cancers exhibited specific driver mutations/rearrangements. 

Our device not only resulted in a high yield of CTCs but also resulted in capture of heterogeneous CTC subpopulations. Of recovered CTCs, on average, only 31% of CTCs expressed some level of EpCAM, emphasizing the need for label-free approaches for studying lung CTCs. In addition, an average of 45% of CTCs expressed Vimentin. Among 23 analyzed patient samples, 22 samples had Vimentin+ CTCs, indicating the presence of Vimentin+ EMT-like cells in circulation in NSCLC. We monitored CTC counts in patients over the course of their therapy and found that the counts mirrored patient clinical outcomes.

In addition to single CTCs, we have observed CTC clusters (≥2 CTCs) in 96% of NSCLC patients. Patients who had more CTC clusters had worse PFS. Among the total number of clusters (2754 clusters per mL) recovered from all 23 patient samples, 75% of CTC clusters were negative for EpCAM expression, whereas 41% of CTC clusters expressed the EMT marker, Vimentin, suggestive of an EMT phenotype in CTC clusters. These findings indicate the advantage of the label-free approach for isolating CTC subpopulations in both single and cluster forms.

We have been able to detect tumor-related mutations in CTCs of patients with driver mutant lung adenocarcinoma. We have shown that recovered CTCs from selected patients with *ROS1*, *ALK*, and *RET* rearrangements carried genomic aberrations matching the primary tumor. 

Although in vivo applications of label-free technologies are still very limited, the capability of collecting recovered CTCs/clusters from the device using a continuous processing technique while in a suspension state opens the opportunities not only for CTC expansion off chip, but also for ex vivo drug testing to direct patient-specific therapies.

## Figures and Tables

**Figure 1 cancers-12-00127-f001:**
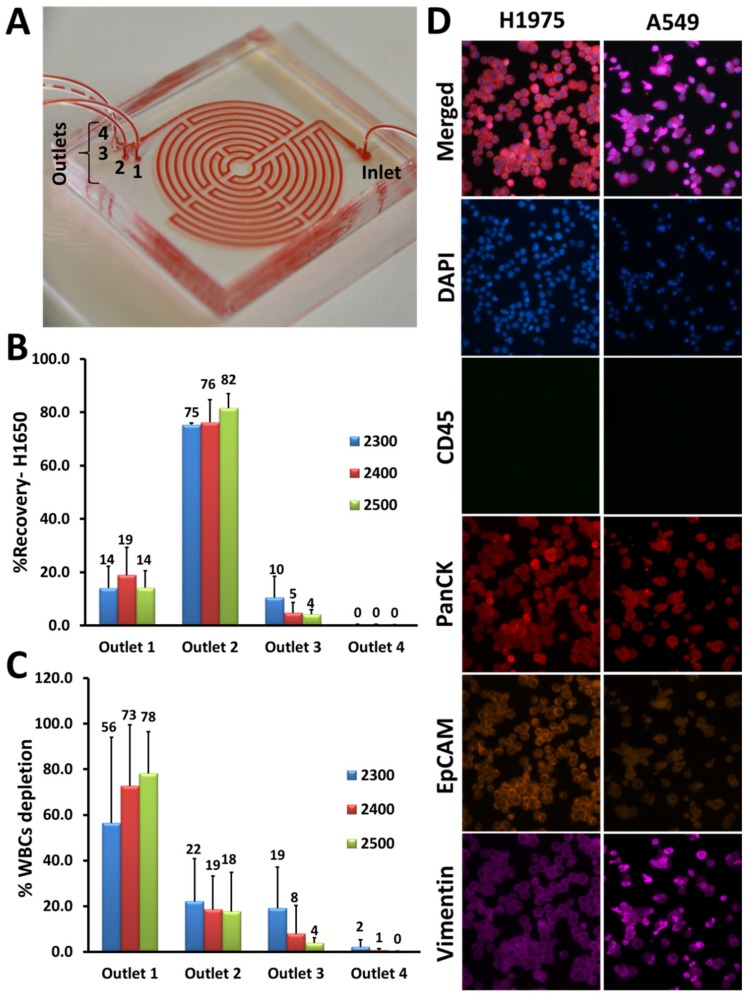
Optimization of Labyrinth for cell recovery. (**A**) The Labyrinth was loaded with red dye to show the device’s structure as well as the inlet and outlets. (**B**,**C**) Using the H1650 lung cancer cell line, different flow rates ranging from 2300–2500 μL/min were tested for inertial separation of cancer cells. Pre-labeled H1650 cell line and DAPI (4′,6-diamidino-2-phenylindole)-labeled white blood cells (WBCs) (1000 cells) were spiked into PBS and processed through the Labyrinth. Using a flow rate of 2500 μL/min, 82% ± 5% of H1650 cells were recovered from outlet #2 and 78% ± 18% of WBCs were removed through outlet #1. (**D**) Immunofluorescence staining optimization. Anti-human CD45 (cluster of differentiation 45) (green), anti-human PanCK (pan-Cytokeratin) (red), anti-human EpCAM (Epithelial cell adhesion molecule) (orange), and anti-human Vimentin (pink) antibodies were tested with lung cancer cell lines, H1975 and A549.

**Figure 2 cancers-12-00127-f002:**
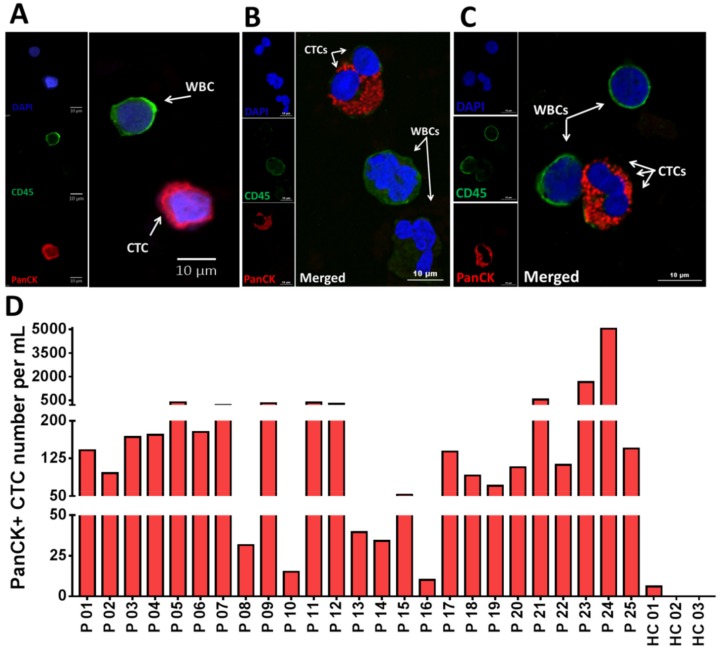
Isolation of circulating tumor cells (CTCs) from non-small-cell lung cancer (NSCLC) patients (*n* = 25). (**A**) Fluorescent microscope image of a single CTC. Cells are stained with DAPI (blue), PanCK (red) and CD45 (green). (**B**,**C**) Confocal microscopy images of some CTC clusters. (**D**) An individual bar plot of the number of CTCs recovered from NSCLC patient samples at baseline, using Labyrinth. The overall number of CTCs in NSCLC patient samples was 417 ± 1023 per mL, while healthy controls had 1 ± 1.7 CTCs per mL.

**Figure 3 cancers-12-00127-f003:**
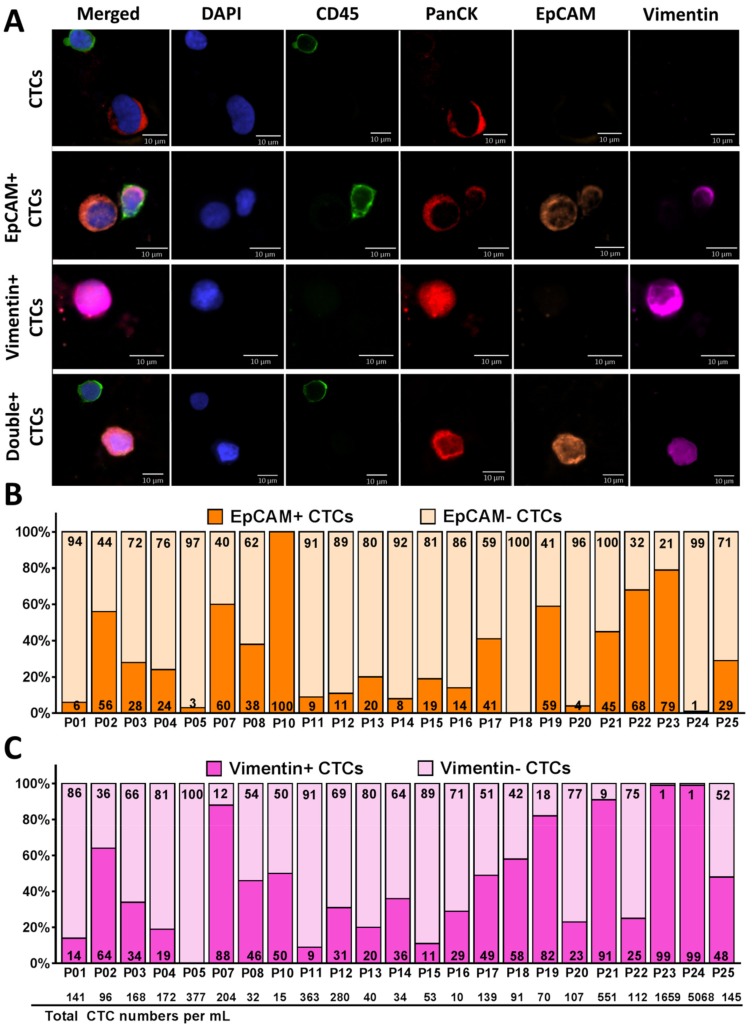
Identification of heterogeneous CTC subpopulations isolated from NSCLC patient samples (*n* = 23). (**A**) Fluorescent microscope images of different subpopulations of CTCs (CTCs, EpCAM+ CTCs, Vimentin+ CTCs, and Double+ CTCs). Cells are stained with DAPI (blue), CD45 (green), PanCK (red), EpCAM (orange), and Vimentin (pink). (**B**,**C**) The percentage of CTCs expressing both EpCAM (**B**) and Vimentin (**C**) recovered from each NSCLC patient sample (*n* = 23). The EpCAM+/− CTCs is shown in dark/light orange respectively and the Vimentin+/− CTCs is shown in dark/light pink respectively. An average of 31% of the captured CTCs were EpCAM+ and 69% were EpCAM− CTCs. An average of 45% of the captured CTCs were Vimenin+ and 55% were Vimentin− CTCs. The total number of CTCs/mL across all patient samples is shown on the bottom of the graph.

**Figure 4 cancers-12-00127-f004:**
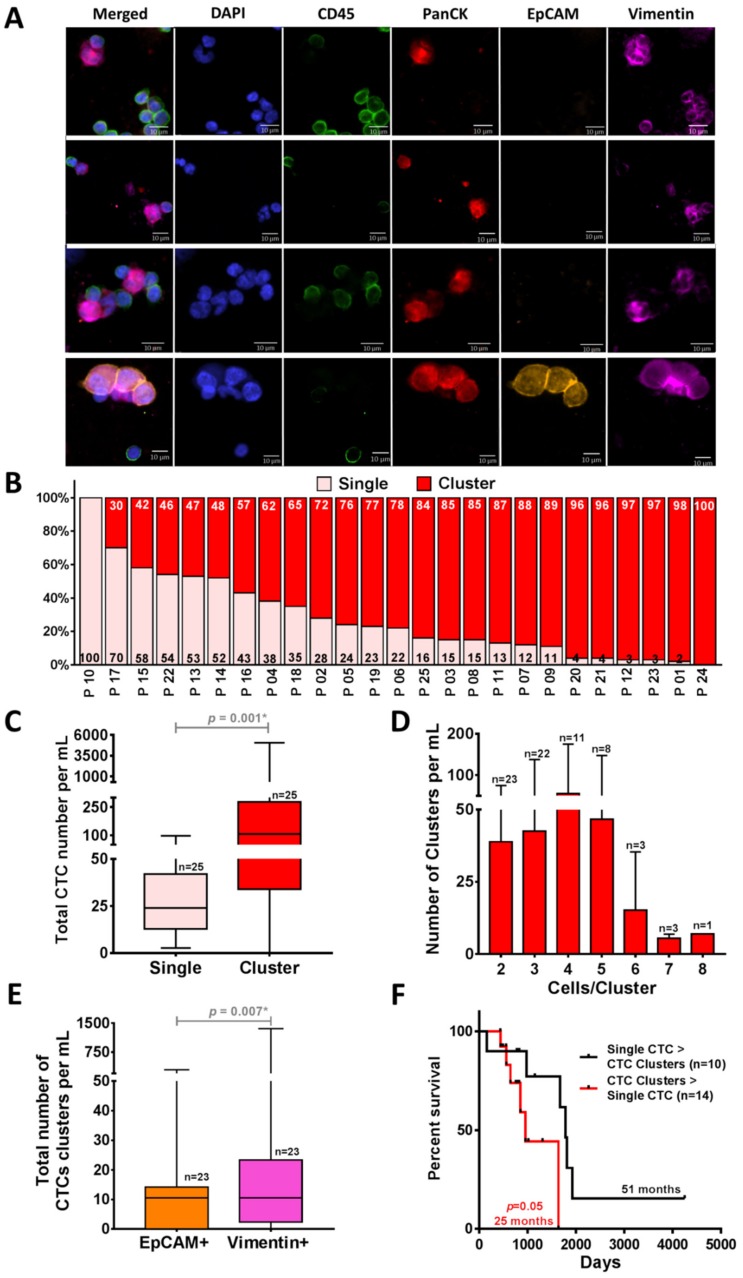
Isolation of CTC clusters from NSCLC patient samples (*n* = 25) by using Labyrinth. (**A**) Immunofluorescence staining images of the representative recovered CTC clusters in NCSLC patients. CTC clusters are stained with DAPI (blue), CD45 (green), PanCK (red), EpCAM (orange), and Vimentin (pink). (**B**) The percentage of CTCs in single (light pink) vs. cluster (red) forms. Across all patients, only one patient did not have CTC clusters. (**C**) Comparison between the total CTC numbers in single and cluster forms. Significantly higher numbers of clusters compared to the single CTCs were observed in the captured CTCs from NSCLC patients (*n* = 25) (*p* = 0.001). (**D**) Cell clusters of 2–8 CTCs were observed in 96% of patients. (**E**) Of the recovered CTC clusters, 41% displayed a mesenchymal or epithelial-to-mesenchymal transition (EMT) phenotype (97 CTC clusters/mL) (*p* = 0.007). Wilcoxon test analysis was used for comparing single vs. clusters CTCs and EpCAM+ vs. Vimentin+ CTCs. Analyses were conducted using GraphPad Prism. (**F**) Comparison of Kaplan–Meier progression-free survival (PFS) graph in patient samples (*n* = 24) with a higher number of clusters than single CTCs (*n* = 10) (red) and in patients who had a higher number of single CTCs (*n* = 14) (black). A higher number of CTC clusters than the single CTCs was correlated with worse PFS (*p* = 0.05). Log-rank (Mantel–Cox) tests were used to analyze the Kaplan–Meier PFS.

**Figure 5 cancers-12-00127-f005:**
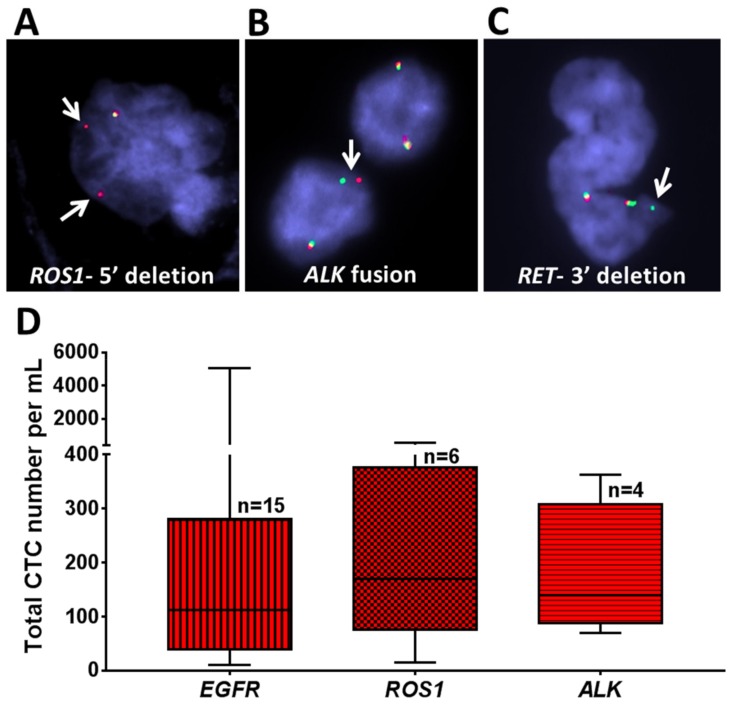
Genomic analysis of NSCLC patient samples using fluorescence in situ hybridization (FISH) analysis. (**A**–**C**) Recovered CTCs from selected patient samples with different mutations (*ROS1*, *ALK*, and *RET*) were evaluated. (**A**) A patient with *ROS1* rearrangement showed 5′ deletion in some of the cells. (**B**) A patient with aberration in *ALK* showed *ALK* fusion. (**C**) A patient with aberration in *RET* showed 3′ deletion. Arrows indicate specific aberration in each gene. (**D**) Box plot of the total number of CTCs recovered from patients with different mutations (*EGFR* (*n* = 15), *ROS1* (*n* = 6), and *ALK* (*n* = 4)). Analyses were conducted using GraphPad Prism.

**Figure 6 cancers-12-00127-f006:**
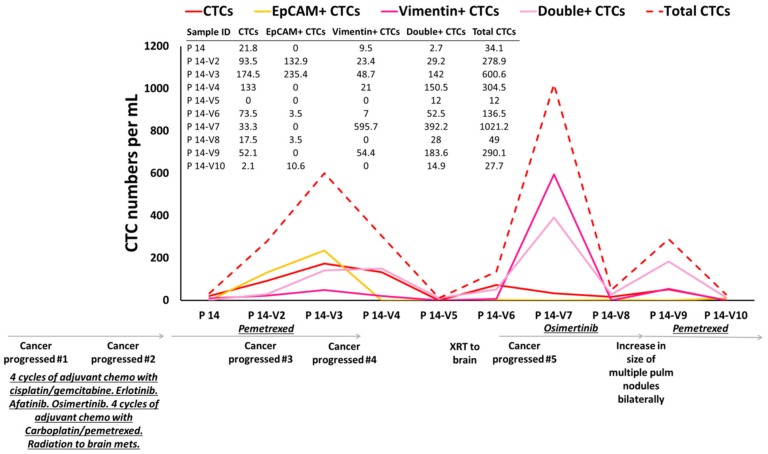
CTC enumeration of one NSCLC patient over the treatment regime. In total, 10 different follow-up samples from patient P 14 were collected and the CTC numbers were evaluated over different treatments. According to the available clinical information, the CTC numbers tracked the patient’s outcome over the course of treatment.

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
