# Peer review of "High-Throughput Label-Free Isolation of Heterogeneous Circulating Tumor Cells and CTC Clusters from Non-Small-Cell Lung Cancer Patients"

_cancers, 2020, doi:10.3390/cancers12010127_

Round 1

Reviewer 1 Report

The authors have used their inertial microfluidic device, called Labyrinth, to detect CTCs and CTC clusters in NSCLS patients. While their findings seem promising, their manuscript needs to improve significantly before being accepted for publications. Some recommendations to improve the quality of this work are as follows:

1) The novelty of the work is not clear. There are many papers in CTC separation using inertial microfluidics since 2008, mainly after Prof. Di Carlo's work. The authors need to thoroughly refer to those works esp. with Spiral microfluidics and explain the added advantages of their Labyrinth device.

2) Using inertial microfluidics at relatively high flow rates seems problematic in detecting CTC clusters. I wonder how the authors have managed to detect the clusters of CTCs without being dissociated. They need to discuss this issue and explain the applicable Reynolds number to detect CTCs vs CTC clusters.

3) It seems that only three values of flow rate, 2300, 2400 and 2500 ul/min, have been tested prior to clinically evaluate the performance of the device. The results in Figure 1B-C show that 2500 ul/min leads to better detection. The authors need to use some analytical or numerical investigation to show that they have engineered their device based on the CTC size (or if probably they have done that in their previous works, they can refer the readers to their own papers for such analysis). One may wonder what range of Reynolds number is applicable in their device, or why they did not try higher flow rate, say 3000 ul/min.

4) The results of detecting CTCs from NSCLS patients shown in Figure 2 and Figure 3C seem unrealistic. For example, they have detected over 5000 CTCs in 1 ml of the blood sample of Patient 24! It is surprisingly high values of CTCs in the blood. Even more than 10 CTCs in 1 ml of blood, even in the late stage of metastatic cancers, seem too many. The authors need to carefully address this issue and compare their work with other relavant clinical dates in detecting the range of CTCs in NSCLS patients. 

5) They need to add the fabrication technology of their inertial microfluidics device in the Material and Method section.

6) Finally, the manuscript needs to professionally proofread as it is full of typos and grammatical errors. Also, the figure numbers are wrong as the following errors  keep appearing throughout the manuscript:

“Error! Reference source not found”

Reviewer 2 Report

In this manuscript the authors report the performance of a microfluidic device that, after been tested on breast and pancreatic cancer patients in a previous publication, is now applied in the metastatic NSCLC setting.

Some comments:

Introduction: authors don't consider that their microfluidic technology might loose a relevant number of single CTC (for example those of smaller dimensions or abnormal plasticity).

Cell-culture paragraph: performance of the test are not detailed. Moreover, presented results may be affected by the condition in which spike experiments have been performed, that seems to be very distant to real samples condition.

Patients' sample paragraph: the amount of blood used s not present. Again, number of cells and contaminants is not clearly described.

Supplementary files are not present.

"Error! Reference source not found". Have the authors checked the manuscript before submission?

It seems that 1 patient has been lost in the Kaplan-Meier curve.

Any information about patients is missing.

Discussion:

The authors don't mention ctDNA testing, that is an established, predictive, FDA approved test for detection of EGFR mutations in not biopsable NSCLC patients.

Abstract:

Since the authors report that the comparison between patients with CTC clusters and those with single CTCs, in terms of PFS, is not significant, it should be removed.

Round 2

Reviewer 1 Report

The authors have carefully addressed my comments and improved the quality of the their manuscript in the revised version. 

Although I still believe that the reported numbers of CTCs in some patients are surprisingly high, the results of this study and the clinical data can be appealing for a broad range of audiences. As such, I recommend this work for publication at Cancer journal after addressing the following minor issues:

The manuscript still needs to be proofread, preferably by a native, competent English writer. Also, there are few acronyms that need to be defined in the first place, such as: NGS, FMSA. ctDNA (it was first introduced in Line 92 but defined in full term in Line 237), etc.

Reviewer 2 Report

All points have been addressed, thanks